# Luminescent Yb^3+^,Er^3+^-Doped α-La(IO_3_)_3_ Nanocrystals for Neuronal Network Bio-Imaging and Nanothermometry

**DOI:** 10.3390/nano11020479

**Published:** 2021-02-13

**Authors:** Géraldine Dantelle, Valérie Reita, Cécile Delacour

**Affiliations:** Institut Néel, University Grenoble Alpes, CNRS, Grenoble INP, 38000 Grenoble, France; valerie.reita@neel.cnrs.fr

**Keywords:** luminescence, harmonic iodate nanocrystals, neurons, biophotonics, nanothermometry

## Abstract

Dual-light emitting Yb^3+^,Er^3+^-codoped α-La(IO_3_)_3_ nanocrystals, known to exhibit both second harmonic signal and photoluminescence (PL), are evaluated as optical nanoprobes and thermal sensors using both conventional microscopes and a more sophisticated micro-PL setup. When loaded in cortical and hippocampal neurons for a few hours at a concentration of 0.01 mg/mL, a visible PL signal arising from the nanocrystals can be clearly detected using an epifluorescent conventional microscope, enabling to localize the nanocrystals along the stained neurons and to record PL variation with temperature of 0.5% K^−1^. No signal of cytotoxicity, associated with the presence of nanocrystals, is observed during the few hours of the experiment. Alternatively, a micro-PL setup can be used to discriminate the different PL lines. From ratiometric PL measurements, a relative thermal sensitivity of 1.2% K^−1^ was measured.

## 1. Introduction

Temperature is a robust parameter that describes and predicts the state of many systems [1]. In neuronal networks, it is strongly linked with the metabolism and state of cells, triggering biochemical reactions, activation rates, and thermal sensitive ionic channels. Slight changes of temperature can affect the dynamics and the reactivity of molecular processes, or reveal disfunction and abnormal developments leading to pathological states. At sub-cellular scales, the expected changes of temperature associated with cellular functions or disfunction is expected to be very weak (100 mW, 10^−5^ K) and/or localized (100–1 nm) which makes the measure highly challenging [2]. Such small variations could remain hidden and screened by the surrounding extracellular solution which exhibits high thermal conductivity, close to water, around 0.6 Wm K^−1^ [3].

In order to detect subtle temperature differences in neurons, one needs non-invasive, local, stable measurements and highly sensitive thermosensors around the metabolism temperature (37 ± 3 °C). The use of luminescent nanoprobes, whose emission properties are dependent on temperature, can be the solution for intracellular thermometry as it is still less invasive and can provide high spatiotemporal resolution [4,5]. Luminescent nanoprobes can be internalized by neurons or adsorbed on their membranes and be in close contact with the reactive sites, providing a method to sense local and weak variations of temperature along many individual cells at the same time [6,7]. 

There are many ways to measure temperature through photoluminescence (PL), monitoring either the excited state lifetime, the luminescence intensity, the variation of emission peak position, etc. [8]. Several thermal nanoprobes have been investigated, such as fluorescent proteins [9], gold metal nanoclusters [10], nanodiamonds [6], polymer [11] or quantum dots [12], carbon nanodots [13], and perovskite nanocrystals [14]. Among them, lanthanide-doped inorganic nanocrystals are of particular interest for their narrow emission lines, related to the f-f orbitals of lanthanide ions involved in the optical transitions and allowing spectral discrimination, for their relatively long excited state lifetime allowing autofluorescence removal by time-gated experiments and for their numerous emission lines covering the visible and near-infrared range [15]. With these nanoprobes, typical luminescence-based nanothermometry consists of a ratiometric approach, consisting in measuring the luminescence intensity ratio (*LIR*) of two emission peaks that are in thermal equilibrium. To compare the thermal performances of different types of nanoprobes, a relative thermal sensitivity, *S_r_*, has been defined as [16]:(1)$$S_{r}=\frac{1}{LIR}\;\frac{\partial LIR}{\partial T}$$

Many Nd^3+^-doped nanocrystals (SrF_2_ [17], LaF_3_ [18], NaGdF_4_ [19], CaF_2_ [20], NaHoF_4_ [21], YAG [22]) have been developed for in vivo imaging as Nd^3+^ present emission lines in the biological transparency windows situated in the near-infrared range (680–1700 nm), limiting autofluorescence from the tissues [23,24,25]. However, these Nd^3+^-doped single-band nanothermometers present a moderate *S_r_* (ca. 0.1–0.5% K^−1^, depending on the host matrix [26]), leading to a relatively high temperature uncertainty *δT*, of the order of 2 °C at 37 °C [18,27,28] with:(2)$$\delta T=\frac{\delta LIR}{LIR}\;\frac{1}{S_{r}}$$
with *δLIR*, the standard deviation of the *LIR*.

For in vitro experiments, for which visible-emitting nanoprobes can be used, Er^3+^,Yb^3+^-doped nanocrystals present a good alternative as they offer higher *S_r_* values (>0.2% K^−1^), reducing consequently the uncertainty to less than 1 °C and even 0.1 °C in some matrices and in some more complex core-shell architectures [15,29,30]. The high performances of Er^3+^-based nanothermometers have been very recently confirmed theoretically by Suta and Meijerink, who demonstrated that Er^3+^ ions fulfil all requirements for optimum performances (high *S_r_* in the physiological temperature range) [31].

Yb^3+^,Er^3+^-doped α-La(IO_3_)_3_ nanocrystals were recently developed with the goal of proposing multi-functional nanoprobes for in vitro imaging. The key advantage of these nanocrystals lies in their dual emission properties [32,33]: (1) As non-centrosymetric nanostructures, non-linear optical signals, such as the second harmonic, can be generated upon near-infrared femto-second excitation [34]; and (2) the facile, thus efficient, substitution of La^3+^ by Er^3+^ and Yb^3+^ ions leads to PL properties in the visible and near-infrared range [35]. Such a combination paves the way to bio-imaging using multiple and complementary imaging techniques: Fluorescence imaging using a conventional epifluorescent microscope and in-depth imaging using a two-photon microscope equipped with a short-pulse excitation laser. We report here an additional feature of these Yb^3+^,Er^3+^-doped α-La(IO_3_)_3_ nanocrystals and demonstrate, as a proof of principle, their capacity to be used as nanothermometers in neuronal cells. After the measurement of the thermal sensitivity in various environments (powder, liquids), the nanocrystals are loaded in cortical and hippocampal neurons for in vitro bioimaging and temperature measurements.

## 2. Materials and Methods

**Nanocrystal synthesis.** Yb^3+^,Er^3+^ doped α-La(IO_3_)_3_ nanocrystals (La_0.895_Yb_0.1_Er_0.005_(IO_3_)_3_) synthesis. LaCl_3_∙6H_2_O, ErCl_3_∙6H_2_O, and YbCl_3_∙6H_2_O (Strem Chemicals, Bischheim, France, purity > 99.9%) were mixed, in stoichiometric proportions, in water (10 mL) with a total concentration of 0.05 mol/L. Separately, HIO_3_ (Sigma Aldrich, Saint Quentin Fallavier, France, purity > 99.5%) was also dissolved in water, with a molar concentration of 0.15 mol/L. Both solutions were mixed together and poured into a quartz vessel, then placed into an Anton-Paar Multiwave PRO microwave oven (Anton-Paar, Graz, Austria). The heating was conducted at 250 °C for 10 min to crystallize the α-La(IO_3_)_3_ phase [35]. After cooling down, the white precipitate is retrieved by filtering and washed several times with distilled water before drying at room temperature for 24 h.

**Nanocrystal structural characterizations.** The powder X-ray diffraction (XRD) pattern was recorded on a Siemens D8 Advance diffractometer (λ_Cu_ = 1.54056 Å) (Brucker, Germany) at room temperature. Scans were collected in the 2Ɵ range 16–90°, with a step of 0.005° and 1 s of acquisition time. Transmission electron microscopy (TEM) images were recorded using a Philips CM300 microscope (Eindhoven, Netherlands), operating at 300 kV and equipped with a TemCam F416 TVIPS camera. Due to the lack of stability of the iodate nanocrystals under the electron beam, the experiment was performed using a liquid-nitrogen cooled sample holder. The experiment at lower temperature (estimated between 77 and 170 K) allows preserving the nanocrystals from being modified under the beam.

**Photoluminescence measurements.** PL and PL excitation spectra of powdered Yb^3+^,Er^3+^ doped α-La(IO_3_)_3_ nanocrystals were measured using a commercial Safas fluorimeter (Xenius, Monaco) at room temperature. Using a tunable vertically-polarized femtosecond laser (Insight X3 Spectra-Physics, 80 MHz, 100–120 fs, Newport, Evry, France) as the excitation source, the non-linear properties of these nanocrystals dispersed in solution were recorded, as detailed in [33]. Temperature-dependent PL measurements were performed using a micro-PL setup coupled with a temperature stage (Linkam THMS600, Scientific Instruments, Tadworth, UK). The excitation was performed using a CW argon laser (488 nm, typical power: 100 kW.cm^−2^) and focused on the sample through an Olympus objective (×60). The emission was collected through the same microscope objective and detected, after a razor edge filter at 488 nm, through a T6400 spectrometer equipped with a Symphony Camera detector cooled at 77 K. During the cell culture, the photoluminescence is collected in liquids using an upright-style epifluorescent Olympus BX51 microscope. The incident illumination (white light, xenon lamp) is filtered at λ_exc_= 488 ± 20 nm and focused on the sample with a ×60 water-immersion objective. The emitted photons are collected through the same objective and filtered at λ_em_ = 535 ± 25 nm. A high-pass dichroic mirror at λ_dic_ = 510 nm suppresses the contribution of the incident photons on the signals that is recorded with sCMOS camera cooled between 80 and 70 K. 

**Cell culture.** E16.5 pregnant mice were killed by cervical elongation. Hippocampi were dissected from the mouse embryos in HBSS-HEPES (Gibco), incubated for 15 min in trypsin 2.5% (Gibco), rinsed in HBSS-HEPES twice, and mechanically dissociated by repeatedly pipetting in 1 mL attachment medium (MEM supplemented with 10% fetal bovine serum, 1% glutamine, and 0.05% peniciline/streptomycine, Gibco). Cells were seeded with a density of 150,000 cells/mL on the glass coverslip previously sterilized and coated with poly-L-lysine (PLL, 1 mg/mL) to ensure neuron adhesion. Neurons were then incubated at 37 °C and 5% CO_2_ in the attachment medium, and replaced a few hours later by serum-free glial conditioned medium supplemented with 1 mM AraC. 

**Nanocrystals loading.** Nanocrystals were dispersed in 37 °C culture medium with ultrasonic bath to prevent large aggregates, and add to the cells medium at a final concentration ranging from 0.1 to 0.001 mg/mL. Constant agitation was maintained for a few minutes. Then cells with nanocrystals were incubated at 37 °C and 5% CO_2_ until observations (around 3 h). 

**Immunostaining.** After PL acquisition neurons were fixed in 3.9% paraformaldehyde (10 min), permeabilized in PBS-0.1% Triton-X100 (3 min) and blocked in PBS-2% bovine serum albumin (10 min). After each step, the solution (PFA, PBS-Triton, or PBS-BSA) is removed and the cells are rinsed at least three times in PBS. Primary antibody anti-YL1/2 (1:1000, BioRad, Hercules, CA, USA) is diluted in PBS-2% BSA (1:1000), incubated with cells for 1 h. Then the solution is replaced by FITC-conjugated secondary antibodies (Alexa Fluor, dilution 1/500) for 1 h. Lastly, cells are then incubated with DAPI (1:1000, Sigma-Aldrich, France) to visualize cells body. The two stainings (YL1/2 and DAPI) enable labelling the microtubule cytoskeleton and the soma respectively to locate the neurites and the cells body. Finally, cells were washed again in PBS, and mounted in Mowiol (Sigma-Aldrich) mounting medium between two coverslips. Optical and immunofluorescence images were collected using either an Olympus BX51(Tokyo, Japan) or a Zeiss AxioImager M2 microscope (Aachen, Germany).

## 3. Results

### 3.1. Temperature Dependent PL of Nanocrystals

Yb^3+^,Er^3+^-doped α-La(IO_3_)_3_ nanocrystals (with doping concentrations of 0.5 mol% Er^3+^ and 10 mol% Yb^3+^) were prepared by a microwave-assisted hydrothermal method as reported in [35]. The obtained nanocrystals, imaged by low-temperature transmission electron microscopy (TEM, Figure 1a), present an average size of 40 nm and a good polydispersity (±10 nm), obtained by the size analysis on more than one hundred nanocrystals. The PL and PL excitation properties of powdered nanocrystals were recorded in the visible range and are presented on Figure 1b. The main excitation peak is situated at 380 nm, corresponding to the ^4^I_15/2_ → ^4^G_11/2_ transition of Er^3+^. There are two intense emission peaks upon a 380 nm excitation, situated at 522 nm and 544 nm, corresponding to the ^2^H_11/2_ → ^4^I_15/2_ and ^4^S_3/2_ → ^4^I_15/2_ transitions of Er^3+^, respectively. A small PL peak is observed around 660 nm, corresponding to the ^4^F_9/2_ → ^4^I_15/2_ transition of Er^3+^. This weak intensity attests to a low population rate of the emitting level (^4^F_9/2_). As this level is populated either from non-radiative de-excitations from the ^4^S_3/2_ level or through Er^3+^-Er^3+^ cross-relaxation mechanisms, the latter being very unlikely at this low Er^3+^ concentration (0.5 mol%) [36,37], it shows that non-radiative mechanisms are not dominating in the considered nanocrystals. Under a 1064-nm femto-second excitation, these nanocrystals exhibit a second harmonic signal (Figure 1c).

Using a micro-PL setup, the emission spectrum of Yb^3+^,Er^3+^-doped α-La(IO_3_)_3_ nanocrystals dried on a silicon substrate was recorded in the 500–560 nm range. Due to the much better spectral resolution of this setup, compared to the one of the fluorimeter, the sub-structure of the emission peaks, whose maximal intensity lies at 522 nm and 544 nm, can be visualized (Figure 2a), corresponding to the transitions from the multiple Stark sublevels of the ^2^H_11/2_ and ^4^S_3/2_ levels to the Stark sublevels of the ^4^I_15/2_ ground state of Er^3+^. 

Thanks to a temperature stage (Linkam THMS600, Linkam Scientific Instruments, Tadworth, UK) coupled to the micro-PL setup, PL spectra were recorded at different temperatures (varying between 25 and 70 °C) after a 10 min thermalization at each temperature. When temperature increases, the intensity of the two emission peaks at 522 nm and at 544 nm, respectively labelled *I*_522_ and *I*_544_, tends to equalize, as the two emitting levels, ^2^H_11/2_ and ^4^S_3/2_, are in close proximity in energy and their population is ruled by Boltzmann equation [38,39]: (3)$$\frac{I_{522}}{I_{544}}=LIR=A\;\exp\left(-\frac{\Delta E}{k_{B}T}\right)$$
with *A* is a constant depending on the host matrix, ∆*E* the energy difference between the ^4^S_3/2_ and ^2^H_11/2_ levels, *k_B_* the Boltzmann constant, and *T* the absolute temperature. The evolution of *LIR*, plotted in a logarithmic scale, as a function of *1*/*T*, follows a linear evolution in agreement with Boltzmann theory (Figure 2b). The slope allows for the calculation of ∆*E* ~730 cm^−1^. This value can be compared to the value calculated from the PL spectrum, taking into account the barycenter of both peaks at 522 and 544 nm, situated respectively at 19,160 cm^−1^ and 18,410 cm^−1^, which gives a gap between the ^2^H_11/2_ and ^4^S_3/2_ levels of 750 cm^−1^, in good agreement with the ∆*E* value obtained from the nanothermometry measurements.

In order to compare the thermal sensitivity of these Yb^3+^,Er^3+^-doped α-La(IO_3_)_3_ nanocrystals with other luminescent nanoprobes, their relative thermal sensitivity, *S_r_* was calculated from Equation (1). As the ∂LIR/∂T term is the derivative of the function described in Equation (3), *S_r_* can be further evaluated by [40,41]: (4)$$S_{r}=\;\frac{\Delta E}{k_{B}T^{2}}$$

At 25 °C, the *S_r_* value for our system is 1.2% K^−1^, which is comparable the values obtained in other Yb^3+^,Er^3+^ doped oxide or fluoride nanocrystals [15,42]. Under those experimental conditions, the temperature uncertainty, *δT*, is evaluated at ~0.5 °C.

The same experiment was repeated on nanocrystals dispersed in two different liquid environments: in ethylene glycol, which is known to stabilize these nanocrystals [32], and in water at pH 7 for further prospects in biology. Once the nanocrystals have sedimented onto the silicon substrate, their emission was recorded as a function of temperature and the LIR value can be studied as a function of temperature (Figure 2b). Like in powder, the linear evolution of *LIR* with *1*/*T* attests from a Boltzmann distribution of the two excited state levels. One can note that the value of *LIR* varies according to the environment, which can be explained by a change of thermal conductivity at the nanocrystals interface. Recent works report the key role of the environment and interfaces of nanothermometry: Bastos et al. evidence a significant variation of *S_r_* values for nanocrystals dispersed in water and in D_2_O [43]. It should be specified that all our experiments were performed using a micro-PL set-up involving high excitation power density (~100 kW cm^−2^) at 488 nm, where heating of the liquid medium is very likely. In order to assess the reliability of these nanocrystals as nanothermometers, further work would be required, measuring PL dependence according to the environment (pH, nature of the tissues, etc.) and the operating conditions (excitation power densities, excitation wavelength, etc.) [44].

### 3.2. Impact of the Nanocrystals on Cultured Neurons

As previously described [45], cortical and hippocampal neurons are extracted from mouse embryos (E16.5) (detailed in Material and Methods). Cells are plated on glass coverslips previously coated with cell adhesive polymers (poly-L-lysine at 1 mg/mL), and then incubated at 37 °C and 5% CO_2_. Then nanocrystals are loaded several days of culture (DIV4 or 5) and a few hours before the recordings. Depending on the incubation time, nanocrystals will penetrate the plasmid membrane or remain adsorbed on neuron membranes. For each condition, Yb^3+^,Er^3+^-doped α-La(IO_3_)_3_ nanocrystals are dispersed in water, at high concentration (0.1 mg/mL) using ultrasonication (stock solution). Then, small amounts of the nanocrystal suspension (5–50 µL) are added to the cell medium under constant agitation for 10 min. Finally, cells are incubated before the observations to recover. 

We adjusted the concentration of nanocrystal, ranging from 0.01 to 0.001 mg/mL. A high concentration of 0.1 mg/mL appears lethal for neurons, while a low concentration significantly reduces the probability to attach nanocrystals on neurites. The final concentration was fixed at 0.01 mg/mL, which enables to keep healthy neurons and several sensing sites where nanocrystals and neurites are colocalized. Most of the large aggregates are removed during media changes (Figure 3a). The remaining large crystals provide controls for measuring PL of the nanocrystals, while smaller nanocrystals are rather internalized or attached on the membranes of neurons (Figure 3a–d).

Figure 3 illustrates the typical shape of five-day old neurons incubated 3–4 h with nanocrystals. The neurons appear healthy despite the presence of nanocrystals, with the expected cell density, number of neurites per cell and neurite length for this maturation stage. The spreading of neurites over all the substrate indicates also the healthy state of cells. No neurite bundles nor large soma aggregates are observed which are first signs of stress or cytotoxic effects within in vitro neural networks. Additionally, there is no sign of membrane disruption. 

### 3.3. Temperature Dependence of the Nanocrystals PL in Neuronal Environment

We assess here the ability to monitor the PL of the nanocrystals in standard conditions of cells observation using an epifluorescent Olympus BX51 microscope, equipped with a FITC-filter cube (excitation 480 nm ± 20 nm; dichroic mirror 505LP; emission filter 535 ± 25 nm). Nanocrystals (0.01 mg/mL) were seeded on bare coverslips (bath #1) and on neurons cultured five days on glass coverslips (batch #2). To record the PL signals, we used a high-magnification fluorescent objective (×60) immersed in the suspension containing the nanocrystals and the neurons. The temperature of the solution is varied from room temperature 20 °C to 40 °C or 45 °C, with and without neurons, respectively, by using the same temperature stage (Linkam THMS600) than the one used on the micro-PL setup. Samples are maintained at the desired temperature for 5 min before the measurement. Emitted photons are collected during 50 ms with a sCMOS camera (Andor Zhyla 5.5) for each temperature steps, while keeping constant all other parameters such the incident illumination (6 mW), the grey scale, or the dynamic range (16 bits). The total intensities emitted from the nanocrystals are then extracted with ImageJ software (ImageJ 1.x, LOCI, University of Wisconsin, USA). 

PL of the Yb^3+^,Er^3+^-doped α-La(IO_3_)_3_ nanocrystals is clearly observed on the optical micrograph as illustrated within Figure 3. The transmitted wavelengths of the FITC cube ranges from 510 to 550 nm (>80%) with a maximum at 525 nm (95%), and enables collecting the main emission spectra of the nanocrystals with two main emission peaks at 522 and 544 nm (Figure 1). Large aggregates identified in the dark field (dark clusters within Figure 3a) correlated with fluorescent cluster on the FITC micrograph, as the one highlighted by the arrows gives an example (Figure 3b). Additionally, smaller crystals that are barely seen in visible wavelengths appear clearly in the FITC range, characterized by small fluorescent dots along the neurites and the somas that can be distinguished from the microtubule filaments stained with YL1/2-FITC markers (closer views, Figure 3c,d). 

Regarding the PL dependence with temperature, we observe a change of PL intensity when increasing the temperature from 20 °C to 45 °C. This change is reversible when decreasing the temperature. Without cells (red dots, Figure 4) in liquid, the nanocrystal PL exhibits a rather linear dependence with the temperature, varying of about 15% from room temperature to 45 °C with a maximal variation of the PL intensity around 0.5% K^−1^ (given by the slope of the red curve, Figure 4). This value is close to the variation measured in micro-PL spectroscopy: ∂LIR/∂T~0.65% K^−1^ (Figure 2). The slightly lower sensitivity obtained with the epifluorescence microscope is expected as we cannot distinguish the contribution of the two PL peaks of the nanocrystals, but only the resulting contribution of both. The contribution of the first PL peak (522 nm) is the most sensitive to temperature change and corresponds the maximum of the transmission of the FITC filter. However, its variations may be hidden by the second peak which exhibits higher PL signals (almost twice). 

In the presence of cells, the PL intensity still decreases when increasing the temperature and the observed variations remain in the expected ranges of intensity, with a maximum sensitivity around 37 °C. However, the recorded variations are weaker and the dependence with the temperature is unclear around 25 °C. At the wavelengths of interest (in the range of 488–550 nm), neurons are not optically transparent, they can either absorb or scatter the optical signals. In particular, the scattering coefficient is high in the visible range [46] and could impact the collected intensity in a significant manner, which may explain the non-linear dependence with the temperatures that we observed. Additionally, cells could act as a thermal barrier between the regulated substrate and the nanocrystals, making slightly less efficient the thermal coupling between the bath and the temperature stage. In both cases, such effects may contribute to increase the non-linearities on the recorded PL signals and make more difficult the extraction of the PL signal of interest around room temperature. These possible artefacts can be overcome using the PL intensity ratio over the absolute PL intensity variation that lies in more sensitive to temperature change. 

To enhance the thermal sensitivity and repeatability we could isolate the contribution of the two nanocrystals PL peaks at 522 and 544 nm. This could be obtained by adding dichroic mirrors on a second emission line to select the first and/or the second peak only. Its contribution can then be isolated and subtracted from the first emission line to improve the signal-to-noise ratio. 

As an alternative, we assessed the possibility to measure temperature of a neuronal network with the micro-PL setup (previously described). An aqueous solution of Yb^3+^,Er^3+^-doped α-La(IO_3_)_3_ nanocrystals (c = 0.1 mg/mL) was dispersed onto the neuronal network (Figure 5a). Once the nanocrystals have sedimented, the excitation beam is focused onto a small nanocrystal aggregate situated onto a dendrite (circled in black in Figure 5a) in order to record the PL spectra at different temperatures in the physiological range. The PL signal of the small aggregates is easily detectable despite the increase of the luminescence background, attributed to the microscope cover slide. In agreement with the measurements made using the conventional microscope, the overall PL intensity decreases with the temperature increase. The *LIR* varies linearly with temperature, in a similar manner as for the nanocrystals in water (Figure 5b), in agreement with previous experiments using quantum dots as thermal sensors [7]; however, the overall accuracy is decreased due to the presence of a background signal due to the glass coverslip, leading to a temperature uncertainty, *δT* of 2 °C. This value could be improved by using a different sample container.

## 4. Discussion

In this work, we demonstrate the possibility to use two different optical microscopy techniques to obtain a local temperature measurement within cultured neuron networks: fluorescent-based imaging and micro-PL spectroscopy. This includes possible observations with an epi-fluorescent microscope mounted with a conventional FITC-filter, which makes the technology ready to use in cell culture laboratories. Fluorescence-based imaging is a promising strategy for cell thermometry because it can be non-invasive and can have a high spatial resolution (diffraction limited level, around 200 nm) on a wide field of view (mm²). When associated with highly sensitive luminescent thermometers, it could enable following very small temperature changes along single neurons and large networks. As luminescent nanoprobes, we investigate here Yb^3+^,Er^3+^-doped La(IO_3_)_3_. Many rare-earth-doped nanocrystals are known as Boltzmann-based nanothermometers, whose temperature variation is monitored by following the PL intensity ratio of two emission lines of Er^3+^. Though developed a few decades again, many challenges still remain for their use in biology, including the control of the nanocrystal size, morphology, stability, optical efficiency, and thermal sensitivity. 

Under a conventional microscope, the temperature-dependent PL intensity of the nanoprobes deposited on a glass substrate follows a typical Boltzmann distribution (Figure 4, red data), which is as expected for temperature-sensitive fluorescent materials. Nonradiative processes increases with the temperature, while radiative transition rate remains constant which results in a decrease of the emission from the excited state [47,48]. Inversely, the PL intensity increases when reducing the temperature. As a result, the chronograms of PL intensity provide a direct, local, and fast temperature measurement (between 0.2 s and 2 ms depending on the PL intensity) over millimeter scale network areas. In comparison, quantum dots based on red-shift emission require multiple acquisition of the PL spectrum which are usually too long (typ. 0.5–1 s) and limited to one specific location per spectrum, preventing larger scale thermal mapping [49]. 

We demonstrated here that Yb^3+^,Er^3+^-doped α-La(IO_3_)_3_ nanocrystals can either enter into neuronal cells or remain adsorbed on cell membrane, enabling dual extra and intracellular measurements over physiological temperature range 20–40 °C with a maximum sensitivity at 37 °C. The variation of PL intensity over temperature that we reached (0.5% K^−1^) is enough to capture thermal difference of few degrees. These thermal gradients are expected within cell compartments [11,50] or during neurostimulation [51,52] to give a few examples. However further enhancements would be required to reveal weaker changes (10^−4^–10^−5^ K) associated with cell activity [2,53]. 

Although there are numerous papers reporting the thermal performances of nanothermometers [15], fewer of them deal with in vitro experiments [54]. From the ratiometric approach on the micro-PL setup, we measured a relative thermal sensitive *S_r_* of Yb^3+^,Er^3+^-doped α-La(IO_3_)_3_ nanocrystals of 1.2% K^−1^ in neuronal cells, higher than previously reported values for other inorganic materials, including Green Fluorescent Proteins (GFP) [55], Nd^3+^-doped nanocrystals (0.1% K) [18], or quantum dots [56] (Table 1). This value is comparable to that of other Er^3+^-doped materials [57,58] or hybrid materials [59], as a consequence of the energy gap between the two emitting levels of Er^3+^ (^4^S_3/2_ and ^2^H_11/2_) whose splitting does not differ much with the host matrix. More complex systems, such as the association of rare-earth doped nanocrystals with chromophores presenting a triplet-triplet annihilation mechanism, can provide higher sensitivity (7% K^−1^) and high thermal resolution (0.1 K) [60].

Scattering and absorption may be non-negligible for neuronal sensing especially when the number of collected photons are low, as it could impact both the number of collected photons or cell metabolism. Although this effect should be less within in vitro cultured neurons than for in vivo applications, it could impact the measurements and the temperature dependence of the collected PL intensity. Future works should investigate how absorption and scattering by neurons could introduce significant artefacts on the measurements, as well as the stability of the measurement depending on the position of the emitters within the cell. Extraction of the PL depth-profile within the cellular layer might be a way to perform such analysis that would be required for in vivo and in vitro applications.

In addition, future work will investigate the gain that could be reached using a conventional microscope with dedicated set of emission/excitation filters to capture separately the emission of the two excited states at 525 nm and at 542 nm, as using the PL intensity ratio over the absolute PL intensity variation lies in more sensitive to temperature change. Indeed, it allows to get rid of the fluctuations of the experimental conditions (excitation beam power, mechanical drift, etc.) and thus reducing possible measurement artefacts.

For all these reasons, Yb^3+^,Er^3+^-doped La(IO_3_)_3_ nanocrystals developed here appear as suitable candidates for contactless (luminescent) thermometers. It is worth noting that, in this work, Yb^3+^,Er^3+^-doped α-La(IO_3_)_3_ nanocrystals were excited at 488 nm, i.e., directly in an excited level of Er^3+^. However, to envision future experiments, including in vivo experiments, which require an excitation in a transparency window of the biological tissue (i.e., in the near infrared), it will be possible to excite these nanocrystals at 980 nm, i.e., in the Yb^3+^ ions, which already prove to efficiently transfer their energy to Er^3+^ via up-conversion processes.

## 5. Conclusions

Amongst the different nanoprobes that are developed for nanomedicine applications, one of the most desired features is their versatility, which allows for their use under different conditions and/or environment and for different purposes (e.g., diagnostics, therapy). Yb^3+^,Er^3+^-doped α-La(IO_3_)_3_ nanocrystals already exhibit two different functionalities: the second harmonic generation and the photoluminescence, which are rarely obtained at the nanoscale. These properties enable visualizing them using conventional fluorescence microscopes, commonly available in biology, and also with non-linear microscopes for in-depth imaging. In addition to this dual-light emission, we show here that these nanocrystals, exhibiting a relative thermal sensitivity of 1.2% K^−1^ and a PL intensity variation with temperature of 0.5% K^−1^, appear as promising candidates for in vitro bio-imaging and thermal sensing of neuronal cells. Indeed, they combine both a high spatial resolution and large field of view and enable measurements over physiological temperature range (20–40 °C) with epi-fluorescent microscopy which makes the technique ready to use in cell culture laboratories. 

## Figures and Tables

**Figure 1 nanomaterials-11-00479-f001:**
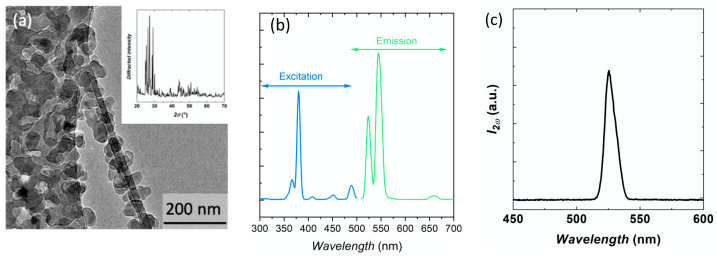
(**a**) Low-temperature TEM image of Yb^3+^,Er^3+^-doped α-La(IO_3_)_3_ nanocrystals. The inset shows the powder X-ray diffraction pattern associated with these nanocrystals. (**b**) Room-temperature PL and PL excitation spectra of Yb^3+^,Er^3+^-doped α-La(IO_3_)_3_ nanocrystals. The PL spectrum was recorded under a 380 nm excitation, whereas the PL excitation was recorded while monitoring the emission at 545 nm spectrum. (**c**) Second harmonic scattering signal from Yb^3+^,Er^3+^-doped α-La(IO_3_)_3_ nanocrystals in ethylene glycol under a 1064-nm excitation.

**Figure 2 nanomaterials-11-00479-f002:**
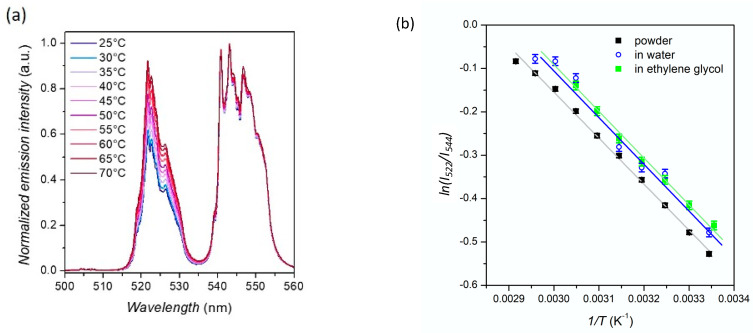
(**a**) Normalized PL spectrum of Yb^3+^,Er^3+^-doped α-La(IO_3_)_3_ nanocrystals dried on a silicon wafer, normalized at 544 nm, as a function of temperature (λ_exc_ = 488 nm, 100 kW∙cm^−2^). (**b**) Evolution of the *LIR* = *I*_522_/*I*_544_ ratio with temperature, for Yb^3+^,Er^3+^-doped α-La(IO_3_)_3_ nanocrystals in different media.

**Figure 3 nanomaterials-11-00479-f003:**
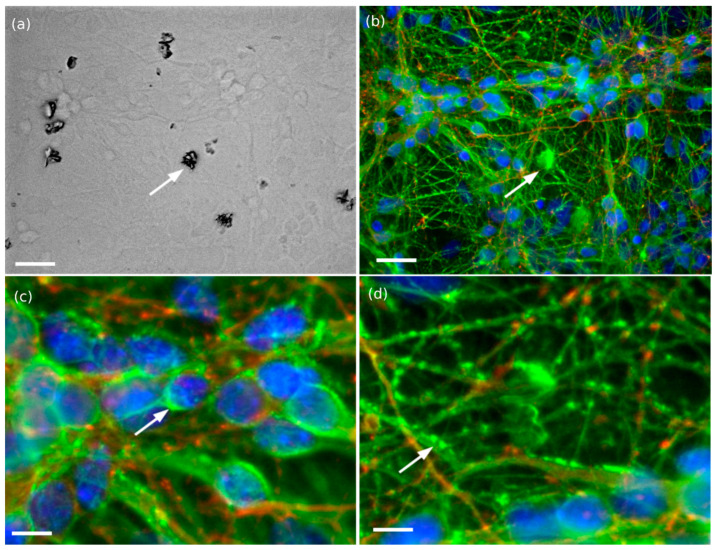
Bright field and fluorescent micrographs of hippocampal neurons (after five days in culture) loaded with Yb^3+^,Er^3+^-doped α-La(IO_3_)_3_ nanocrystals (concentration of 0.01 mg/mL). Neurons are stained with DAPI (blue), YL-1/2-FITC (green) and synapsin-TRITC (red) to locate the soma, neurites, and synapses. FITC fluorescent signals of Yb^3+^,Er^3+^-doped α-La(IO_3_)_3_ nanocrystals and microtubules are superimposed (λ_exc_ = 488 nm, λ_em_ = 525 ± 15 nm, 6 mW). We can clearly distinguish the large clusters of nanocrystals by comparing optical and fluorescent images (**a**,**b**), as well as smaller nanocrystals along the somas (**c**) and the neurites (**d**) as highlighted with the arrows. Scales are (**a**,**b**) 30 µm, (**c**,**d**) 10 µm.

**Figure 4 nanomaterials-11-00479-f004:**
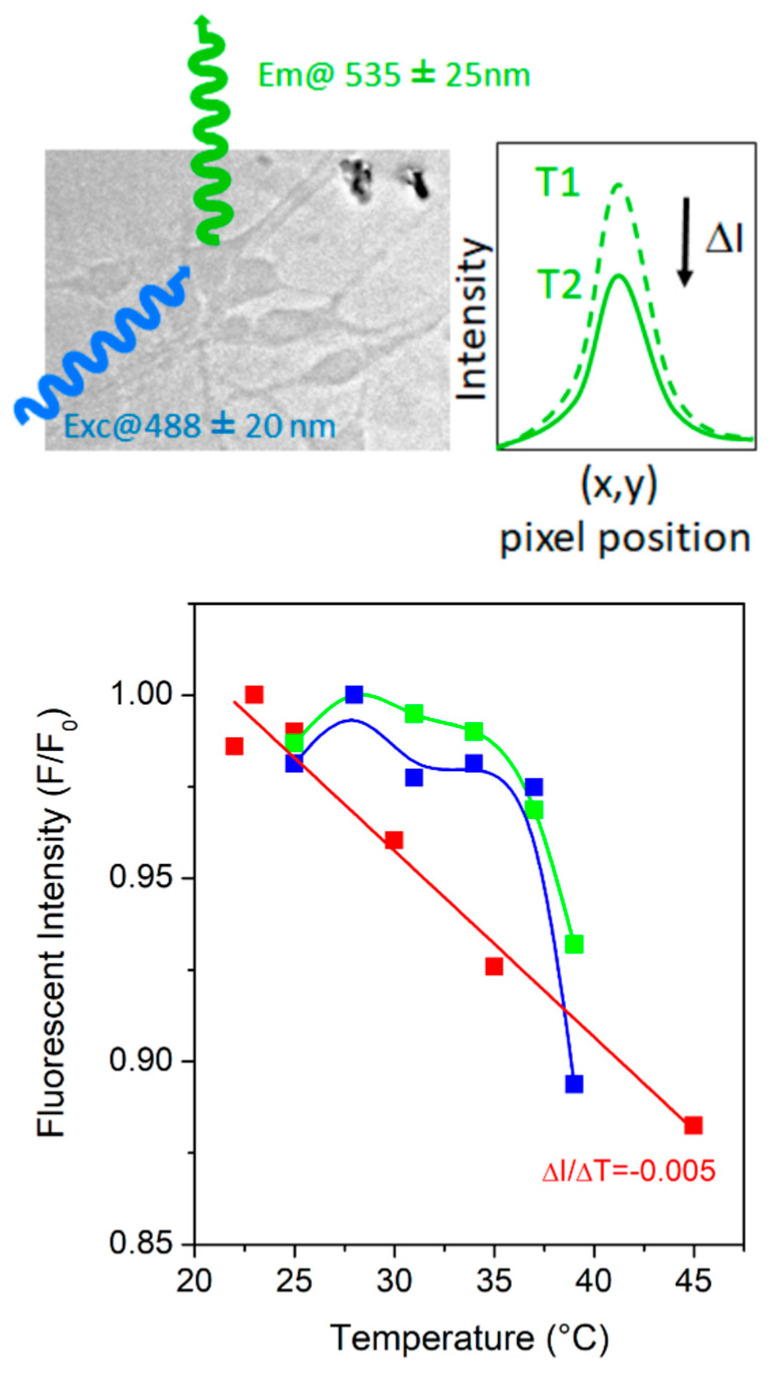
Temperature dependence of the PL of Yb^3+^,Er^3+^-doped α-La(IO_3_)_3_ nanocrystals monitored with an epifluorescent microscope (Olympus BX51), without and with neurons (red and green/blue dots, respectively). The intensity per pixel is collected with a ×60 water-immersion objective, as a function of the substrate temperature adjusted with a Linkam controller. A FlTC-filter cube enables to select the incident illumination (xenon lamp) at the nanocrystal excitation wavelength λ_exc_ = 488 ± 20 nm and to collect their emission around λ_em_ = 535 ± 25 nm. A high-pass dichroic mirror at λ_dic_ = 510 nm suppresses the contribution of the incident photons on the recorded signals. The signal is monitored with sCMOS Andor Zhyla 5.5 camera at a constant exposure time 50 ms.

**Figure 5 nanomaterials-11-00479-f005:**
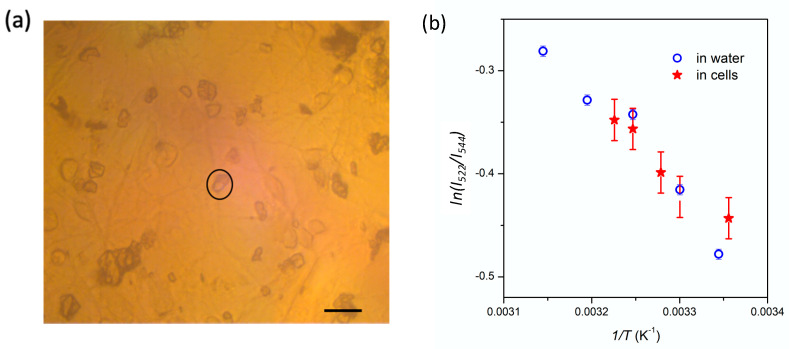
(**a**) Optical image of Yb^3+^,Er^3+^-doped α-La(IO_3_)_3_ nanocrystals loaded onto neuronal cells with a concentration of 0.1 mg/mL, in water (Scale bar: 12 µm). (**b**) Evolution of LIR of the nanocrystals adsorbed on neurons as a function of temperature.

**Table 1 nanomaterials-11-00479-t001:** Comparison of relative thermal sensitivity (*S_r_*) and thermal resolution (*δT*) of different fluorescent thermal sensors used in cells.

Samples, Ref.	*S_r_* (% K^−1^)	*δT* (K)	Type of Cells
La(IO_3_)_3_:Er^3+^,Yb^3+ †^	1.2	<1	Cultured neurons
Fluorescent proteins [55,61]	0.4	0.5	Cultured HeLa cells
NaYF_4_:Nd^3+^ [18]	0.1	<1	HeLa cancer cells
Quantum dots [56]	1.0	<1	Cancer cells
NaYF_4_:Er^3+^,Yb^3+^ [58]	1.6	<1	HeLa cancer cells
Luminescent nanogels [62]	NP	0.3–0.5	Living COS7 cells

^†^ this work. NP: not provided.

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
