# Peer review of "Luminescent Yb3+,Er3+-Doped α-La(IO3)3 Nanocrystals for Neuronal Network Bio-Imaging and Nanothermometry"

_nanomaterials, 2021, doi:10.3390/nano11020479_

Round 1

Reviewer 1 Report

This paper reports on the investigation of Ln-doped nanocrystals as luminescent nanothermometers. Typical figures of merits, the relative sensitivity and the temperature uncertainty, are measured on powders. As a proof of principle, nanocrystals are also incubated with neuronal cells to test their ability of detecting temperature in a real system.

In my opinion this paper tackles an interesting subject and could be published after these points have been addressed.

At lines 96-104, two different setups are described. However, at lines 234-236, it is said that a FTIC filter cube is used: I guess in this case the fluorescence signal does not go through the T6400 spectrograph as mentioned before. I would suggest to better explain the setups in the Material and methods section.

Equation 3. The ratio is calculated between intensities at 522 and 540 nm. Why has it been chosen 540, where the fluorescence band rises very steeply and not for example a point near the maxima or minima of the band centred at about 545 nm? Or the bands individually integrated?

In section 3.2, the fluorescence is measured summing up all the signal that goes through the filter cube and it is correlated to temperature in Figure 4.

However, in my opinion, working in this way, the self-absorption can be an issue: depending on the point in which the laser is focused, the sample may present different absorption / extinction of both the laser and the fluorescence (tissues are not transparent in the range 488-550 nm): hence, even if the temperature is homogeneous in the sample, different fluorescent signals will be measured. Self-absorption would not be an issue if a ratio of two closely lying bands is used.

Can the authors comment on self-absorption, explaining if their method is reliable?

Reviewer 2 Report

Géraldine Dantelle and colleagues proposed Yb3+, Er3+-doped α-La(IO3)nanocrystal as novel luminescent materials for nanothermometry applied to neuronal structures. Although the data are very promising, there are some issues that need an explanation in order to better focus the further practical use of such luminescent tracers for neurons.

My comments are:

  • The neuron culture conditions and nanocrystal addition have to be better explained. For example, it is not clear if the cell culture medium is removed before the acquisition of fluorescent micrograph or not. How could the authors understand if nanocrystals are simply deposited on top of neurons or if they are internalized into neuronal cells?
  • It is not clear the reason of using FITC derivative to marker neurites since it has the emission in the same region of dots. How could the authors clearly distinguish between different contributions? From fluorescent images it is only possible to identify few bright green dots over neurites. Instead of using markers for neuronal internal structures, could it possible to use only fluorescent nanocrystals?
  • In figure 3-a) large aggregates are visible. If such large aggregates (around 10 µm) are really made of nanocrystals as stated in the text, their emission should be much brighter than that shown in figure 3-b). They seem to be impurities of cell culture medium, covered by fluorescent nanocrystals. Do the authors have some experimental evidence to confirm this hypothesis?
  • Do the authors expect to have different results regarding the temperature dependence if nanocrystals are deposited or internalized by cells? In my opinion, this is crucial for the interpretation of the results in order to use these tracers for the measurement of neuronal temperature in different biochemical conditions.
  • Figure 2-b shows the behavior of fluorescence intensity of nanocrystals in water and in ethylene glycol after their sedimentation in silicon substrate. If possible, this graph should be done also with nanocrystals dispersed in both solvents using a spectrofluorometer equipped with a Peltier attachment to control the solution temperature. The temperature dependence behavior could be slightly different from nanocrystal deposited in a substrate.
  • In figure 4, I would suggest modifying the green label in “emission range 535±25 nm”. The label FITC emission is a bit misleading for the performed measurements. The graph Intensity vs (x,y) should be better explained since it is not clear which is the meaning of (x,y).
  • In line 321 it is not clear the meaning of spatiotemporal, since in this paper time-resolved analyses are not reported. I would should suggest modifying the term in “spatial”.

I would suggest the publication of the paper in Nanomaterials journal after minor revisions.

Reviewer 3 Report

Dantelle et al reported the synthesis of dual-light emitting Yb3+,Er3+-codoped α-La(IO3)3 nanocrystals. These nanocrystals, exhibiting a relative thermal sensitivity of 1.2 % K-1 and a PL intensity variation with temperature of 0.5 % K-1, appear as promising candidates for in-vitro bio-imaging and thermal sensing of neuronal cells. The manuscript can be accepted for publication after addressing the following questions. 1. The authors mentioned “Second Harmonic Generation” for several times. But I don’t find any results on Second Harmonic Generation. 2. More structural characterizations, such as XRD, XPS, HRTEM should be added to confirm the structure of the nanocrystals. 3. TEM was conducted at low temperature, why? What is the temperature? 4. Some relevant references regarding typical luminescent nanothermometry should be added.[ ChemNanoMat 2016, 2 (3), 171-175; J. Mater. Chem. C, 2021, 10.1039/D0TC05056C] 5. More discussion on the temperature dependent PL mechanism is required. “Nonradiative processes increases with the temperature, while radiative transition rate remains constant which results in a decrease of the emission from the excited state.” It is better to add temperature dependent PL lifetime to support this point [Biomacromolecules 2013, 14 (6), 2112-2116].

Round 2

Reviewer 3 Report

All the reviewers' comments are well addressed.

Author Response

No specific comments have been made by the reviewer 3 during this 2nd round. It states "All the reviewers' comments are well addressed. "